# Incidence of Herpes Zoster in HIV-Infected Patients Undergoing Antiretroviral Therapy: A Systematic Review and Meta-analysis

**DOI:** 10.3390/jcm10112300

**Published:** 2021-05-25

**Authors:** Han-Chang Ku, Yi-Tseng Tsai, Sriyani-Padmalatha Konara-Mudiyanselage, Yi-Lin Wu, Tsung Yu, Nai-Ying Ko

**Affiliations:** 1An Nan Hospital, China Medical University, Tainan 709, Taiwan; cd4187@gmail.com (H.-C.K.); yttsai13@gmail.com (Y.-T.T.); 2International Doctoral Program in Nursing, Department of Nursing, College of Medicine, National Cheng Kung University, Tainan 70101, Taiwan; pkadawatha@gmail.com (S.-P.K.-M.); lynnewu727@gmail.com (Y.-L.W.); 3Department of Nursing, National Cheng Kung University Hospital, Tainan 704, Taiwan; 4Department of Public Health, College of Medicine, National Cheng Kung University, Tainan 70101, Taiwan; 5Department of Nursing, College of Medicine, National Cheng Kung University, Tainan 70101, Taiwan

**Keywords:** incidence, herpes zoster, HIV-infected patients, highly active antiretroviral therapy, meta-analysis

## Abstract

The incidence of herpes zoster (HZ) in patients infected with HIV is higher than that of the general population. However, the incidence of HZ in HIV patients receiving antiretroviral therapy (ART) remains unclear. This meta-analysis aimed to estimate the pooled incidence rate and risk factors for HZ in the post-ART era. We identified studies assessing the incidence of HZ in the post-ART era between 1 January 2000 and 28 February 2021, from four databases. Pooled risk ratios were calculated from 11 articles using a random-effects model. The heterogeneity of the included trials was evaluated by visually inspecting funnel plots, performing random-effects meta-regression and using I^2^ statistics. Of the 2111 studies screened, we identified 11 studies that were eligible for final inclusion in the systematic review and 8 studies that were eligible for a meta-analysis. The pooled incidence of HZ in the post-ART era (after the introduction of ART in 1997) was 2.30 (95% confidence interval (CI): 1.56–3.05) per 100 person years (PYs). The risks of incidence of HZ among people living with HIV included male sex (AOR: 4.35 (95% CI: 054–2.41)), men who have sex with men (AOR: 1.21 (95% CI: −0.76–1.13)), CD4 count < 200 cells/μL (AOR: 11.59 (95% CI: 0.53–4.38)) and not receiving ART (AOR: 2.89 (95% CI: −0.44–2.56)). The incidence of HZ is substantially lower among HIV infected patients receiving ART than those not receiving ART. Initiating ART immediately after diagnosis to treat all HIV-positive individuals is crucial to minimize the disease burden of HZ.

## 1. Introduction

Herpes zoster (HZ), also known as shingles, is caused by the reactivation of varicella zoster virus (VZV), which remains latent in the sensory ganglia following varicella infection [1]. HZ is a human immunodeficiency virus (HIV)-associated opportunistic infection. HIV infected populations have a 3- to 20-fold higher risk of contracting HZ than HIV seronegative individuals [2,3,4,5,6]. A systematic review and meta-analysis of the HZ incidence in ART-naive patients was 9.4%. During the first year of ART, the incidence was 2.3%, but this reduction was not statistically significant [7].

Even though the introduction of ART has improved the survival of those living with HIV, the incidence of HZ among HIV cohorts remains higher than that of the general population [4,8]. The incidence of HZ in the HIV cohort was 9.3–141 per 1000 patient years (PY)s [8,9,10]; but the incidence of HZ in the general population is only 3–5 per 1000 PYs [11]. The general population lifetime risk for HZ is between 23.8% and 30% and affects approximately one in four people in Europe; however, for those aged >85 years, the risk increases to one in two people [12].

ART was previously associated with reduced risk of herpes zoster and other opportunistic infections and could substantially improve the quality of life among HIV infected patients [7]. The long-term effects of ART on HZ incidence among HIV infected cohorts have been inconsistent across previous studies. A study in 2005 noted that the HZ incidence has not significantly decreased since the advent of ART [9]. Conversely, other prior studies found a reduction in the risk of HZ after the initiation of ART in North America, Europe, Taiwan and Africa [4,8,10,13,14,15,16,17]. Several studies have shown an increase in the HZ incidence during the first to fourth months following ART therapy, which suggests that HZ may be a feature of immune reconstitution inflammatory syndrome (IRIS) [9,18,19].

A better understanding of the risk factors for HZ for the HIV infected population may provide valuable information for health care professionals to identify adult patients at a heightened risk and could help formulate appropriate policies for vaccination strategies for the HIV-infected population [20]. Therefore, the objective of meta-analysis aimed to systematically review studies estimating the incidence and risk factors for HZ in the post-ART era among people living with HIV (PLWH) and discuss implications based on the updated evidence.

## 2. Materials and Methods

### 2.1. Registration

Our study has been registered with the International Prospective Register of Systematic Review Protocol (PROSPERO). The registration ID number was 240601.

### 2.2. Search Strategy and Selection Criteria

This systematic review and meta-analysis focused on the associations between HIV infection, antiretroviral therapy and HZ. This review was conducted in accordance with the Preferred Reporting Items for Systematic Reviews and Meta-analysis guidelines [21]. We searched four English/Chinese databases, including the PubMed, Embase, Cochrane Library and Cumulative Index to Nursing and Allied Health (CINAHL), to identify observational, population-based studies reporting the incidence of HZ between 1 January 2000 and 28 February 2021. We searched for original articles published using Emtree terms, Medical Subject Heading (Mesh) terms and keywords, including “human immunodeficiency virus”, “human immunodeficiency virus infection”, “antiretroviral therapy (ART)”, “highly active antiretroviral therapy (ART)”, “herpes zoster (HZ)”, and “varicella zoster virus (VZV)”.

This study used the following inclusion criteria: (a) studies on HIV/AIDS population, including both genders and those aged ≥15 years; (b) studies that included compare the pre-ART (before ART) group and the post-ART (after ART) group or a post-ART (after ART) that utilized ART, regardless of the type, dose and route of administration; (c) observational or cohort studies; and (d) population-based studies, which was defined as involving all residents within a specific area and in which the patients were representative of that area; (e) presumptive diagnosis of HZ were as a rash through physician’s examination and diagnosis, and conclusive diagnosis were extracted from ICD codes: 53.0–53.9 (ICD-9) and B02.0-B02.9 (ICD-10). In addition, the following exclusion criteria were implemented: (a) studies in which outcomes of the HZ incidence using ART therapy were not reported and (b) studies where it was impossible to extrapolate or evaluate the necessary data from the published results. We considered all peer-reviewed journals and included observational and cohort studies.

Three authors (Ku, Tsai and Sriyani) independently screened the titles, abstracts and full texts of the papers resulting from the search. In the absence of an abstract, the authors reviewed the full text to determine eligibility. In addition, we manually searched for related papers in the reference lists of the included studies to determine additional pertinent studies. We excluded studies that did not report original data (e.g., review articles). If there were conflicting opinions, the study quality score was determined following a joint discussion with the fourth author (Wu).

After extracting all relevant papers (Figure 1), we excluded duplicate studies and studies that did not relate to the topic. Finally, we included 11 reliable studies that used published criteria for diagnosing HZ.

### 2.3. Risk of Bias in Individual Studies and Data Extraction

We extracted various study characteristics from the original included studies. The data extraction methodology followed the meta-analysis of statistics assessment and review instrument created by the CASP critical appraisal checklist for cohort studies. The data extraction was performed by two appraisers independently and was synthesized after discussion. In addition, the effect of ART on the incidence of HZ among HIV infected individuals was abstracted from each study. Furthermore, three blinded reviewers used the inclusion criteria to select eligible papers. In addition, three appraisers (Ku, Tsai and Sriyani) independently reviewed each study using the critical appraisal skills program (CASP) for cohort studies to assess the risk of bias. The CASP for cohort studies scores ranged from 0 to 12 [22]. The total appraisal scores of each study ranged from 8 to 12 (highest quality). If a study had a score below 8 points, it was considered to be a high risk of bias. Then, we rechecked the study quality together. If there were conflicting opinions, the study quality score was determined following a joint discussion with the fourth author. If there was disagreement, the fourth author was contacted to resolve the disagreement. The four reviewers were consulted when necessary, for the qualitative assessment and data extraction. Any disagreement was solved through mutual consensus.

The data extracted included the main author, country, mean age, study period, associated factors and the overall, and post-ART era incidence of HZ per 100 person years (PYs) (Table 1). We collected data on the incidence per 100 PYs with 95% confidence intervals (CIs). We recorded the incidence per 100 PYs with 95% CIs for the overall study period.

### 2.4. Data Analysis

The statistical analysis was conducted using the Comprehensive Meta-Analysis (CMA) software Version 3.0 (Biostat, Englewood, NJ, USA) [23]. We summarized adverse outcomes for the HIV population in the eligible studies and performed a pooled meta-analysis with a random-effects model of the individual cumulative incidence for each outcome with their 95% Confidence Intervals (CI). The incidence rates were obtained from the selected studies. The incidence rate was calculated by dividing the number of HZ events by all HIV infected patients during the study period expressed as 100 PYs of follow-up. If any heterogeneity was observed, the cause of heterogeneity was first analyzed and then subjected to treatment. To assess the heterogeneity between studies and to quantify the importance of inconsistency between the estimation of the incidence rate, we used the I^2^ statistic (values of <25%, 25–75% and >75% represented low, medium and high heterogeneity, respectively). Seven subgroup analyses and five meta-regressions were performed to investigate potential sources of high heterogeneity.

## 3. Results

This section may be divided by subheadings. It should provide a concise and precise description of the experimental results, their interpretation, as well as the experimental conclusions that can be drawn.

### 3.1. Study Characteristics

During the period from 1 January 2000 to 28 February 2021, we identified 2111 records: 221 from PubMed, 1793 from Embase, 23 from Cochrane Library and 74 from CINAHL. After removing duplications and the initial screening, 45 articles were eligible for full-text review (Figure 1).

Eleven studies met the inclusion criteria for the meta-analysis. The 11 studies received a quality score of ≥8, with nine studies receiving a score of >10 out of 12 (Table 1). The 11 studies included 10 cohort studies and 1 clinical observational study. Overall, the study quality was acceptable. These 11 studies included 195,190 total HIV infected participants. The incidence was reported for populations in America (four studies), Europe (four studies), Asia (one study) and Africa (two studies).

The pooled incidence rate of HZ in the 11 studies was 2.30 (95% CI: 1.56–3.05) per 100 person years (PYs) (Figure 2a). Due to high heterogeneity, subgroup analyses were conducted and stratified by gender, income, AIDS history, study observational years, HIV risk factors and CD4 cell count (Table 2). Among the 10 studies, patients with high income (pooled RR, 2.64; 95% CI, 1.62–3.65) had the pooled HZ incidence greater than those with low income (pooled RR, 1.33; 95% CI, −0.56–3.22). Among the 4 studies, patients without AIDS history (pooled RR, 0.60; 95% CI, 0.46–0.72) had the pooled HZ incidence greater than those who had AIDS history (pooled RR, 0.40; 95% CI, 0.28–0.54). Among the 4 studies, patients being heterosexual had the pooled HZ incidence (pooled RR, 0.41; 95% CI, 0.31–0.52) greater than IDUs and MSM. Of the 4 studies, patients with a CD4 count < 200 cells/mm^3^ (pooled RR, 0.78; 95% CI, 0.55–0.91) were at increased risk of HZ compared with those with a CD4 count > 200 cells/mm^3^ (pooled RR, 0.21; 95% CI, 0.06–0.51).

### 3.2. Meta-Regression Analyses of Overall HZ Moderators

We performed several meta-regression analyses, by which samples were segmented by sex, CD4 count level and ART use.

### 3.3. Subgroup Analyses by Sex

In six studies, males and females were analyzed. The subgroup analyses revealed that males were at significantly higher HZ risk than females (Pooled RR = 0.32 (female) and 0.68 (male); 95% CI: 0.20–0.48 (female) and 0.52–0.80 (male)). Larger effect sizes were observed for the HZ incidence for males than for females in the HIV-infected population (Figure 2b).

### 3.4. Subgroup Analyses by CD4 Count Level

As stated, two levels of CD4 count (CD4 count < 200 and CD4 count > 200 cells/mm^3^) were analyzed in these studies. Of the four studies, patients with a CD4 count < 200 cells/mm^3^ (pooled RR, 0.76; 95% CI, 0.47–0.92, *p* < 0.001) were at increased risk of HZ compared with patients with a CD4 count > 200 cells/mm3 (pooled RR, 0.22; 95% CI, 0.06–0.54, *p* < 0.001). A significant effect was noted in that CD4 count < 200 cells/mm^3^ among HIV-infected population that would increase the risk of HZ (Figure 2c).

### 3.5. Subgroup Segmented by ART Use

Among the six studies that enrolled patients in the pre-ART (not receiving ART) and post-ART (receiving ART) era, the subgroup analyses showed that those in the pre-ART era (pooled RR, 0.05 PYs; 95% CI, 0.03–0.11, *p* < 0.001) were at significantly higher HZ risk than those in the post-ART era (pooled RR, 0.02 PYs; 95% CI, 0.01–0.05, *p* < 0.001). The HIV-infected population who was not receiving ART was at higher HZ risk than the HIV-infected population receiving ART (Figure 2d).

### 3.6. Meta-Regression Analysis

We found a high level of heterogeneity in the HZ incidence across the 11 studies (I^2^ = 99.3%) (Figure 2a). Publication bias was analyzed by generating a funnel plot (Figure 3) and using Egger’s test.

Furthermore, significant publication bias existed based on the Egger’s regression analysis *(p* = 0.002). A univariate analysis was conducted to identify significant independent covariates as potential sources of heterogeneity to be included in the meta-regression model. We found that risk factors for HZ incidence included male sex (AOR: 4.35 (95% CI: 0.54–2.41), *p* = 0.002), CD4 count < 200 cells/μL (AOR: 11.59 (95% CI: 0.53–4.38), *p* = 0.013) and not receiving ART (AOR: 2.89 (95% CI: −0.44–2.56), *p* = 0.16) (Table 3).

## 4. Discussion

To the best of our knowledge, this is the first meta-analytic study to investigate the incidence of HZ in HIV patients with ART. The pooled incidence of HZ in the post-ART era was 2.30 (95% CI: 1.56–3.05) per 100 person years (PYs). The incidence risk of HZ among PLWH included male sex (AOR: 4.35 (95% CI: 0.54–2.41)), MSM (AOR: 1.21 (95% CI: −0.76–1.13)), CD4 count < 200 cells/μL (AOR: 11.59 (95% CI: 0.53–4.38)) and not receiving ART (AOR: 2.89 (95% CI: −0.44–2.56)). The incidence of HZ is substantially lower among HIV-infected patients receiving ART than those not receiving ART. Initiating ART immediately after diagnosis and treating all HIV-positive individuals are crucial approaches for minimizing the disease burden of HZ.

In our analysis, there were 5 studies that have the post-ART HZ incidence only, and the other 6 studies compare pre-ART and post-ART HZ incidence. The first phase of ART therapy era was from 1997 to 2006, after which the protocol of ART medications were continuously updated. After 2010, the latest studies mainly focus on the post-ART era. Meta-analysis study required more research data for conducting empirical explanations, increasing the sample size and doing the subgroup analysis. If we exclude the 5 studies, that may increase the research bias and heterogeneity. Finally, early studies may have higher heterogeneity, if there were newer studies, subgroup analysis could be carried out.

Our finding showed that patients with high income had the pooled HZ incidence greater than those with low income. Accumulating evidence suggests responses to HIV that combine individual level interventions with those that address structural or contextual factors (like housing, income etc.) that influence risks, access to care and health outcomes [25]. Socioeconomic status often determines access to HIV treatment. In resource-poor countries, poverty may prevent access to health care and subsequent treatment; while in resource-rich countries, easy access to care, diagnosis and treatment may early diagnosis of HZ for PLWH [26]. The impact of the introduction of ART on the incidence and mortality of HIV–associated opportunistic infections (OIs) has been well documented in high-income countries (HICs). During the first year of ART, the risk of herpes zoster declined to 2.3%. There was a major reduction in risk for most opportunistic infections with ART use in low- and middle- income countries (LMICs), with the greatest effect seen in the first year of ART treatment [7]. Our finding showed that patients without AIDS history had the pooled HZ incidence greater than those who had AIDS history. Clinically, patients without history of AIDS that come to hospitals for treatment of HZ, and AIDS was concurrent diagnosed with HZ. Herpes zoster was considered to be an early manifestation of HIV-1 infection, and even a harbinger of the AIDS as immunosuppressive conditions [27].

Our finding showed that that male sex, MSM, CD4 count < 200 cells/mm^3^, no AIDS history and not receiving ART were significant risk factors for HZ among HIV infected patients. HIV patients are at an increased risk of HZ because of their impaired immunity and the use of immunosuppressive medications. Our study found that the risk of HZ substantially decreased after ART initiation. The World Health Organization recommended initiating ART immediately for all HIV-positive patients regardless of clinical stage or CD4 cell count level [28]. The effect of ART was greater after one year, which is consistent with the findings of other studies showing a progressive time-dependent reduction in risk over the first two to three years of ART [29]. This would be explained by the occurrence of HZ across a wide range of CD4 counts, with less of a protective effect of immune restoration, and because of the rate of IRIS during the first month after ART initiation [7]. The results from another systematic review and meta-analysis of the incidence of opportunistic infection showed that the HZ incidence in ART-naive patients was 9.4%; during the first year of ART, the HZ incidence was 2.3%, but the effect of ART had a non-statistically significant reduction in the HZ incidence [7]. A limitation of that study was that it assessed 14 studies for the HZ incidence, but the countries were from Sub-Saharan Africa (one RCT and seven cohort studies), Asia (four cohort studies) and Latin America (two cohort studies); it did not include European countries. A systematic literature review showed that annual HZ incidence of general population throughout Europe, varying by country from 2.0 to 4.6/1000 person-years with no clearly observed geographic trend [12]. HZ incidence increases with age, and quite drastically after 50 years of age; incidence rates were higher among women than men, and this difference increased with age. In the European (WHO) database, the overall mortality ranged from 0 to >0.07/100,000. The age- and gender-specific HZ mortality rates from the other databases showed that while in younger age groups the HZ mortality rate was higher in males, in older patients the rate was much higher in women [30]. The HZ incidence was higher in HIV-infected population among pre-ART era than general population [4,14]. Conversely, an advantage of our study was that we assessed the geographic incidence of HZ in HIV-infected populations, include the different countries and race. Our study found that HZ risk in the HIV-infected population substantially decreased following ART initiation, and ART was previously associated with a reduction in the risk of herpes zoster and other opportunistic infections [7]. Reactivation of latent varicella zoster virus, partly due to age-related immunosenescence and immunosuppressive conditions, HZ prevention through the early initiation of ART could substantially improve the quality of life of HIV-infected populations.

Our data also found that male sex and CD4 count <200 cells/mm^3^ were significant risk factors for HZ in HIV-infected patients. Prior studies found that male sex was a risk factor for HZ in HIV-infected patients, while other studies found that female sex was associated with an increased risk [4,11,14,15,31]. HZ is a transmitted disease caused by the varicella zoster virus; therefore, HZ incidence, like MSM, is likely to be higher in HIV-infected patients as they are a sexually active population, that are associated with similar risk factors (e.g., sex, partner change rate, condom use). [10,24,32,33]. An increased HZ risk among females has been noted in both HIV-infected and -uninfected populations, most likely due to differences in healthcare-seeking behavior but possibly due to immunologic or hormonal mechanisms or biological response to VZV [4,34]. Our findings are similar to those of other studies, where an increase in CD4 count was a protective factor [4,8,9,10,13,14,15,31]; however, others have found HZ to occur with a wide range of CD4 counts in HIV-infected patients [3]. The highest HZ risk was observed in patients with CD4 counts <200 cells/mm^3^. This degree of immunodeficiency renders patients susceptible to an increased HZ risk. Increasing CD4 counts, reflecting improved immune function, are probably the primary factor influencing the decreased rates of HZ. Currently, there are two vaccines on the market, Zostavax (a live vaccine) and Shingrix (a recombinant zoster vaccine). Recommendation for vaccination depends on HIV infected individuals’ immunity, suggested for CD4+ Cell Counts >200 Cells/mL after ART use. Zostavax is less effective in older individuals and is contraindicated in immunosuppressive conditions (HIV/AIDS), during immunosuppressive drug therapy and pregnancy [35]. A randomized, double-blind, controlled trial in HIV-infected adults with CD4+ cell counts >200 cells/mL virally suppressed on ART, the results showed that two doses of Zostavax in HIV-infected adults suppressed on ART with CD4+ counts ≥200 cells/µL were safe and immunogenic [36]. The Shingrix vaccine was immunogenic and had a clinically acceptable safety profile in HIV-infected adults in London, Berlin, Germany and Belgium [37], but most countries not used in HIV-infected population due to out-of-pocket.

While the incidence of HZ has decreased in HIV-infected adults on ART suppression, it continues to occur, particularly in the setting of IRIS following the initiation of ART [8,9,20,37]. However, in those studies, immune responses were less robust than in our study. Even though all participants in those studies had CD4+ cell counts, plasma HIV RNA was not reported. Our study found a positive impact of ART with increased CD4 counts enabling HIV-positive patients to avoid an initial HZ episode.

Certain limitations should also be acknowledged when interpreting the results from this study. First, because our meta-analysis included observational and cohort studies, there was high heterogeneity due to the inclusion of diverse populations with various follow-up times, variability in the reporting of HZ, the ART regimen and the duration of ART, and a lack of data on important confounders such as baseline CD4 counts. Moreover, this study is vulnerable to bias introduced by methodologic and clinical heterogeneity in the primary studies. Second, this meta-analysis included only articles published in English, which could lead to reporting and language biases. We could not check variability and inconsistency of diagnostic methods used to determine cases. Third, studies were often based on administrative or electronic medical record databases, and misclassification of exposure due to coding errors and unmeasured or inadequate adjustment of confounding factors may have biased the results. Fourth, we wanted to compare the incidence of HZ in HIV-infected patients among post-ART era mainly. All the 11 studies reported the HZ incidence among post-ART era, and only 6 studies compared the HZ incidence with pre-ART and post-ART, so we could not determine the effect of the ART completely.

Nonetheless, this review presents some important and novel findings. First, few studies distinguish between a presumptive and conclusive diagnosis of HZ. Therefore, we used meta-regression to focus on quantifying estimates of heterogeneity and potential biases. Second, this review selected studies from different countries and ethnicities, whereas prior studies have not focused on European countries. Third, the current study presents findings from 2000 to 2021, whereas few studies focused on such long-term effects of ART, other than for unspecified HZ. 

## 5. Conclusions

This systematic review with meta-regression provides a global estimate of the incidence of HZ among the HIV infected population in the post-ART era. The profound effect of ART on the incidence of HIV-related HZ is the key reason for the observed global decline and highlights the continued priority of expanding ART access. HIV infected patients with lower CD4 count have a higher incidence of HZ. As HIV progresses toward becoming a chronic disease and due to aging-induced immunodeficiency, the changing global burden of HZ with HIV during the next decade will require research into interventions to prevent HZ and innovations in the care of HIV patients with HZ. Clinical management of individuals exposed to varicella-zoster virus should take into consideration the type of exposure, evidence of immunity and host-immune status with regard to ability to receive varicella vaccination safely. Physicians and other healthcare providers can target patient education based on their risk factors and improve the uptake of zoster vaccination. This data will be used for health policy makers to determine and prevention programs for people at risk of herpes zoster among those living with HIV.

## Figures and Tables

**Figure 1 jcm-10-02300-f001:**
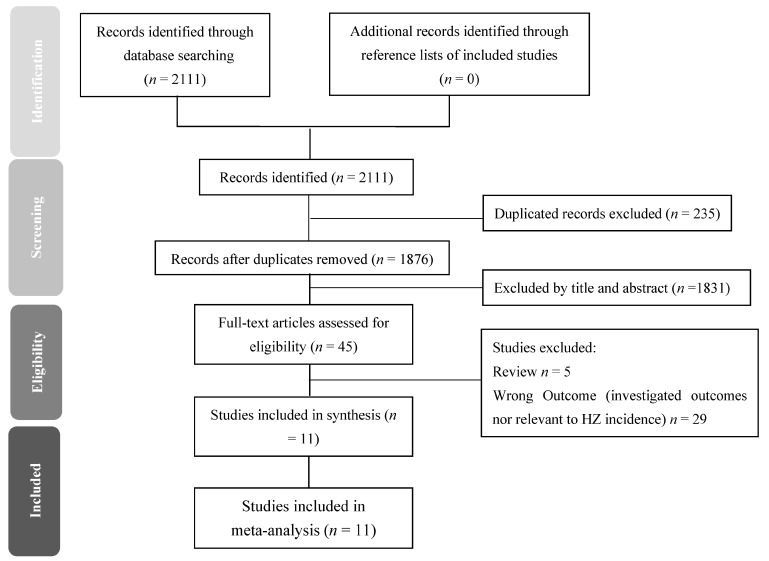
Preferred reporting items for systematic review and meta-analyses (PRISMA) flow diagram for searching and identifying included studies. Database: PubMed (*n* = 221), Embase (*n* = 1793), Cochrane Library (*n* = 23) and CINAHL (*n* = 74). Keywords: (Human Immunodeficiency Virus OR Acquired Immune Deficiency Syndrome Virus* OR Lymphadenopathy-Associated Virus*) AND (HAART OR ART OR Antiretroviral Therapy*) AND (Incidence OR Herpes Zoster OR Varicella Zoster Virus* OR herpes virus infection). Date: available until 28 February 2021.

**Figure 2 jcm-10-02300-f002:**
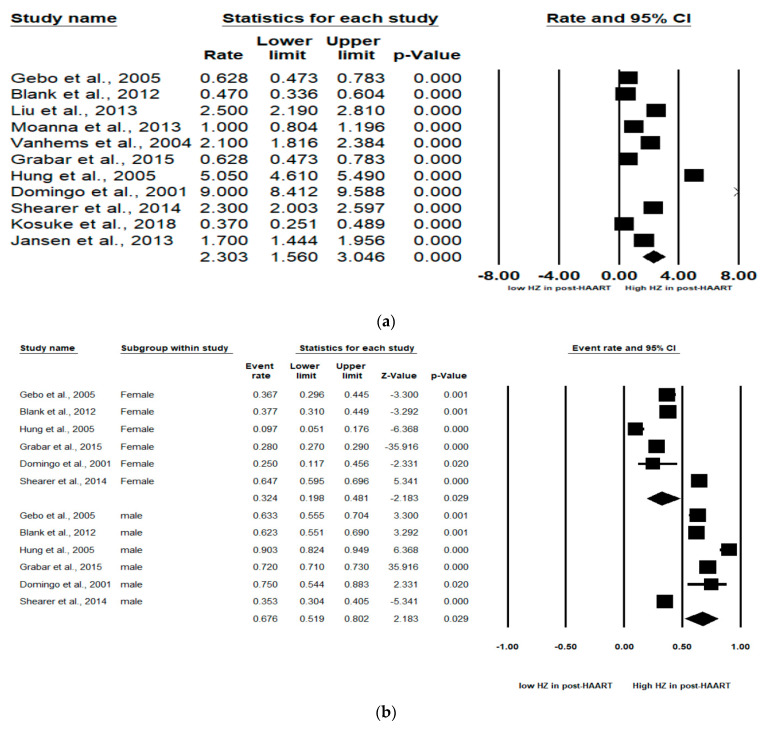
Meta-analyses of (**a**) HZ incidence of HIV-infected patients among ART-era therapy, (**b**) subgroup segmented by sex, (**c**) subgroup segmented by CD4 count level and (**d**) subgroup segmented by ART use. (Note: in the graph, the square represents the effect size of each study. The bigger the square, the more participants in the study. A horizontal line represents the 95% confidence intervals of the study result, with each end of the line representing the boundaries of the confidence interval. The diamond represents the combined effect. Pre-ART = not receiving ART. Post-ART = receiving ART).

**Figure 3 jcm-10-02300-f003:**
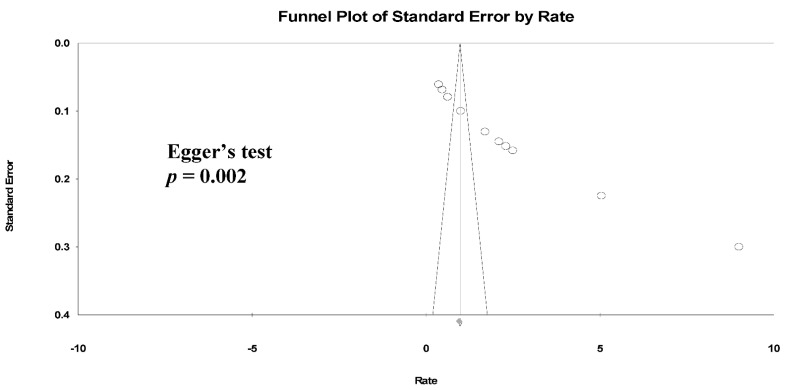
Funnel plots of HZ incidence. (Note: in the graph, the observed studies are shown as open circles and the observed point estimate in log units us shown as a diamond).

**Table 1 jcm-10-02300-t001:** Characteristics of studies reporting the incidence rate of HZ among PLWH (*n* = 11).

Author/Year	Country/ Observational Years	Mean Age (years)	Gender (% men)	Participants	N (*n*/*n*) Without/with ART	Incidence	Associated Factors
Domingo et al., (2001) [18]	Spain/ 2 years	38 (33–43)	75%	316 HIV infected	0/316	9/100 PYs	CD8 count increase
Vanhems et al., (2005) [17]	French/ 18 years	32	88%	441 HIV infected	57/384	2.1/100 PYs	
Hung et al., (2005) [13]	Taiwan/ 9 years	35 (15–83)	91.9%	716 HIV infected	175/541	5.67/100 PYs	CD4 count < 50 cells/mm^3^
Gebo et al., (2005) [9]	U.S.A/ 4 years	41 (23–58)	63%	239 HIV infected	58/100	3.2/100 PYs	ART and CD4 count 50–200 cells/mm^3^
Blank et al., (2012) [8]	U.S.A/ 7 years	39 (32–44)	62%	4353 HIV infected	180/552	0.93/100 PYs	ART within 90 days and CD4 count
Liu et al., (2013) [16]	U.S.A/ 8 years	36	0%	2813 HIV infected	760/2053	2.5/100 PYs	ART
Moanna, Rimland. (2013) [10]	U.S.A/ 29 years	41.5	99%	3816 HIV infected	903/2787	1.0 episodes/100 PYs	CD4 count < 50 cells/mm^3^ and prior HZ history
Jansen et al., (2013) [14]	German/ 25 years	37.8 (14.1–77.3)	58.9%	3757 HIV infected	326/3757	1.7/100 PYs	ART and CD4 count < 100 cells/mm^3^
Shearer et al., (2014) [24]	South Africa/ 7.4 years	36.6	37.8%	15,025 HIV infected	0/15,025	0.74/100 PYs	CD4 count < 50 cells/mm^3^ and prior HZ history
Grabar et al., (2015) [4]	French/ 19 years	40	69.3%	91,044 HIV infected	25,822/65,222	0.63/100PYs	Low CD4 count, female, high HIV RNA levels, low CD4/CD8 ratio and prior AIDS
Kawai et al., (2018) [15]	Tanzania/ 6.8 years	35 (29–41)	28%	72,670 HIV infected	29,342/43,328	0.37/100 PYs	CD4 count < 50 cells/mm^3^, older age, female and year of enrollment

**Table 2 jcm-10-02300-t002:** Subgroup analysis of HZ incidence in different categories.

Subgroup Category	No. of Studies	RR (95% Confidence Interval)	I^2^ (%)	*p* Value
Overall HZ	11	2.30 (1.56–3.05)	99.3	0.001
Sex				
Male	6	0.68 (0.52–0.80)	97.5	0.001
Female	6	0.32 (0.20–0.48)	97.5	0.001
Income				
High income	8	2.64 (1.62–3.65)	99.5	0.001
Low income	2	1.33 (−0.56–3.22)	99.3	0.001
AIDS history				
Yes	4	0.40 (0.28–0.54)	98.3	0.001
No	4	0.60 (0.46–0.72)	98.3	0.001
Observation years				
>7 years	5	2.50 (1.29–3.71)	99.6	0.001
≤7 years	5	2.24 (1.09–3.40)	99.1	0.001
HIV risk factor				
Heterosexual	4	0.41 (0.31–0.52)	76.5	0.005
IDUs	4	0.35 (0.20–0.54)	88.7	0.001
MSM	4	0.32 (0.19–0.49)	90.2	0.001
CD4 count				
CD4 < 200	4	0.78 (0.55–0.91)	95.6	0.001
CD4 > 200	4	0.21 (0.06–0.51)	97.0	0.001
ART use				
Pre-ART	6	6.22 (3.59–8.85)	99.6	0.001
Post-ART	6	2.00 (1.04–2.95)	99.2	0.001

**Table 3 jcm-10-02300-t003:** Meta-regression analysis of HZ risk factors affecting heterogeneity.

Variable	Coefficient	Standard Error	OR (95% CI)	*p* Value
Sex				
Female	Reference			
Male	1.47	0.48	0.54–2.41	0.002 *
HIV risk factor				
Heterosexual	Reference			
IDU	−0.77	0.51	−1.76–0.22	0.13
MSM	0.19	0.48	−0.76–1.13	0.70
CD4 count				
CD4 > 200	Reference			
CD4 < 200	2.45	0.98	0.53–4.38	0.013 *
AIDS history				
No	Reference			
Yes	−0.57	0.42	−1.39–0.25	0.17
ART use				
With ART	Reference			
Absence of ART	1.06	0.77	−0.44–2.56	0.16

* *p* < 0.05.

## Data Availability

No new data were created or analyzed in this study. Data sharing is not applicable to this article.

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
