# Peer review of "Incidence of Herpes Zoster in HIV-Infected Patients Undergoing Antiretroviral Therapy: A Systematic Review and Meta-analysis"

_jcm, 2021, doi:10.3390/jcm10112300_

Round 1

Reviewer 1 Report

The authors performed an interesting systematic review on the incidence of herpes zoster in patients on HAART therapy.

Here are some suggestions for improving the systematic review

  1. How was the risk of Bias measured within the studies?
  2. How was the risk of bias assessed between studies?
  3. It would be advisable to add a chapter on risk of bias assessment.
  4. what are the reasons for such high heterogeneity between studies?
  5. Enter in the Flow chart (figure 1) [(Additional records identified through other sources (n = 0)] the number of records manually searched from the reference lists of the included studies.
  6. Add more information on Figure 3 (funnel PLOt)

Author Response

Response to Reviewer 1 Comments

Point 1: How was the risk of Bias measured within the studies?

Response 1Page 4, Line 138 to 146

“The data extraction was performed by two appraisers independently and was synthesized after discussion. In addition, the effect of ART on the incidence of HZ among HIV infected individuals was abstracted from each study. Furthermore, three blinded reviewers (Ku, Tsai, and Sriyani) used the inclusion criteria to select eligible papers. In addition, three appraisers (Ku, Tsai, and Sriyani) independently reviewed each study using the critical appraisal skills program (CASP) for cohort studies to assess the risk of bias. The CASP for cohort studies scores ranged from 0 to 12 [22]. The total appraisal scores of each study ranged from 8 to 12 (highest quality). “

Point 2: How was the risk of bias assessed between studies?

Response 2Page 4, Line 146 to 152

“If a study had a score below 8 points, it was considered to be a high risk of bias. If there were conflicting opinions, the study quality score was determined following a joint discussion with the fourth author. If there was disagreement, the fourth author was contacted to resolve the disagreement. The four reviewers were consulted when necessary for the qualitative assessment and data extraction. Any disagreement was solved through mutual consensus. “

Point 3: It would be advisable to add a chapter on risk of bias assessment.

Response 3: Page 4-5, Line 135 to 153

“2.3 Risk of bias in individual studies and data extraction

We extracted various study characteristics from the original included studies. The data extraction methodology followed the meta-analysis of statistics assessment and review instrument created by the JBI critical appraisal checklist for cohort studies. The data extraction was performed by two appraisers independently and was synthesized after discussion. In addition, the effect of ART on the incidence of HZ among HIV infected individuals was abstracted from each study. Furthermore, three blinded reviewers used the inclusion criteria to select eligible papers. In addition, three appraisers (Ku, Tsai, and Sriyani) independently reviewed each study using the critical appraisal skills program (CASP) for cohort studies to assess the risk bias. The CASP for cohort studies scores ranged from 0 to 12 [22]. The total appraisal scores of each study ranged from 8 to 12 (highest quality). If a study had a score below 8 points, it was considered to be a high risk of bias. Then, we rechecked the study quality together. If there were conflicting opinions, the study quality score was determined following a joint discussion with the fourth author. If there was disagreement, the fourth author was contacted to resolve the disagreement. The four reviewers were consulted when necessary for the qualitative assessment and data extraction. Any disagreement was solved through mutual consensus.”

Point 4: what are the reasons for such high heterogeneity between studies?

Response 4: 1. I² statistic values of 11 studies are >75%, so we do the subgroup analyses and meta-regression, just like difference income (figure).

Page 5, Line 172 to 174

“Seven subgroup analyses and five meta-regressions were performed to investigate potential sources of high heterogeneity. The statistical analysis was conducted using CMA Software (Version 3.0).”

  1. Because all the 11 studies from different countries, race, sex, clinical or secondary database, so high heterogeneity between studies. And see the first limitation.

Page 10, Line 282 to 287

“A limitation of that study was that it assessed 14 studies for the HZ incidence, but the countries were from Sub-Saharan Africa (one RCT and seven cohort studies), Asia (four cohort studies), and Latin America (two cohort studies); it did not include European countries. Conversely, an advantage of our study was that we assessed the geographic incidence of HZ in HIV-infected populations, include the different countries and race. ”Thank you for your valuable comments. We have revised the paragraph.

Point 5: Enter in the Flow chart (figure 1) [(Additional records identified through other sources (n = 0)] the number of records manually searched from the reference lists of the included studies.

Response 5: Page 3, Figure 1

“Additional records identified through reference lists of included studies other sources (n=0)” 

Point 6: Add more information on Figure 3 (funnel PLOt)

Response 6: Page 8, Figure 3

“Funnel plots of HZ incidence. (Note: in the graph, the observed studies are shown as open circles, and the observed point estimate in log units us shown as a diamond)”

Reviewer 2 Report

This meta-analysis of 11 studies presents an overall incidence rate of herpes zoster in patients with HIV from studies published over the last 20 years. With the recent FDA approval of the Shingrix vaccine and vaccine safety in patients living with HIV (Berkowitz et al. 2015), it is important to understand the risk of herpes zoster in patients living with HIV in the era of available ART. However, the study selection, heterogeneity, and multiple comparisons prevent the paper from drawing its intended conclusions.'

Abstract: 

22: HAART is an antiquated term – recommend changing to ART

29: what is the definition of post-HAART era

Introduction

Line 52 – which population is the stated lifetime risk referring to?

Methods

89: define pre-HAART and post-HAART group - is this temporal (i.e. before and after ART was available) or was this contemporaneous (i.e. patients on and off ART)

89: Inclusion criteria states ‘b) studies that included a pre-HAART group and a post-HAART group.’ Only 6 of these studies compare pre/post-ART incidence ratios. How do the authors determine the effect of ART on the incidence of HZ in the other 5 studies. 

Figure 1 – flow chart is not in correct order in terms of placement of studies excluded due to duplication or abstract

105-110, 138-144 – too repetitive, need to consolidate and streamline the process in which studies were reviewed and agreed upon

Table 1 – column 6 is unclear – are the patients off / on ART or with / without HZ?

Table 1 – does incidence refer to overall or post-HAART incidence of herpes zoster?

Results

187: How do the authors explain their finding that higher income and no history of AIDS are risk factors for herpes zoster?

188: Age is a significant risk factor for herpes zoster and most studies reporting on herpes zoster incidence stratify by age. The authors need to include age in their subgroup analysis.  

Table 2: it is unclear in this subgroup analysis if all subjects are in the post-ART area since all studies include a pre / post ART group by inclusion criteria

Discussion

253: How was this pooled incidence calculated? Does it include only patients on ART from each study? What was the methodology for calculating pooled incidence in the 5 studies that did not compare pre / post ART incidences? How does incidence compare to pre-ART era? 

278: What does this study add by including European countries? What are the expected or hypothesized differences in HZ incidence in this population compared to previous meta-analysis

288: HZ is not caused by herpes simplex virus and is not sexually transmitted

318: How does study distinguish between presumptive and conclusive diagnosis of HZ? Add this to methods section.

Conclusions

What recommendations do you have regarding prevention and interventions and for which populations?

Author Response

Response to Reviewer 2 Comments

Point 1: Abstract:

Line 22: HAART is an antiquated term – recommend changing to ART

Response 1: Page 1, Line 21 to 22 and all the manuscript

“antiretroviral therapy (ART) “

Point 2: Line 29: what is the definition of post-HAART era

Response 2: Page 1, Line 29

The definition of post-ART era is “after the introduction of HAART“

Point 3: Introduction

Line 52 – which population is the stated lifetime risk referring to?

Response 3: Page 2, Line 52 to 53

“The general population lifetime risk for HZ is between 23.8% and 30%, and affects approximately one in four people in Europe “

Point 1:

Response 1:

Point 4: Methods

Line 89: define pre-HAART and post-HAART group - is this temporal (i.e. before and after ART was available) or was this contemporaneous (i.e. patients on and off ART)

Response 4: Page 2, Line 89 to 91

“(b) studies that included a pre-HAART (before ART) group and a post-HAART (after ART) group that utilized HAART, regardless of the type, dose, and route of administration;.”

Point 5: Line 89: Inclusion criteria states ‘b) studies that included a pre-HAART group and a post-HAART group.’ Only 6 of these studies compare pre/post-ART incidence ratios. How do the authors determine the effect of ART on the incidence of HZ in the other 5 studies.

Response 5: Mainly, we want compare the incidence of HZ in HIV-infected patients among post-ART era. So we want to focus the post-ART HZ incidence in 11 studies. Overall, 5 studies reported the post-ART HZ incidence, and only 6 of these studies compare pre-ART and post-ART HZ incidence ratios, and do the subgroup analysis.

Page 11, Line 325 to 329

“Fourth, we wanted to compare the incidence of HZ in HIV-infected patients among post-ART era mainly. All the 11 studies reported the HZ incidence among post-ART era, and only 6 studies compared the HZ incidence with pre-ART and post-ART, so we could not determine the effect of the ART completely.”

Point 6: Figure 1 – flow chart is not in correct order in terms of placement of studies excluded due to duplication or abstract

Response 6: Page 3, Figure 1

“We changed the order of placement of studies excluded due to duplication or abstract.”

Point 1:

Response 1:

Point 7: Line 105-110, 138-144 – too repetitive, need to consolidate and streamline the process in which studies were reviewed and agreed upon

Response 7: Page 3, Line 105 to 108

“After extracting all relevant papers (Figure. 1), we excluded duplicate studies and studies that did not relate to the topic. Finally, we included 11 reliable studies that used published criteria for diagnosing HZ.”

Page 4, Line 142 to 149

“In addition, three appraisers (Ku, Tsai, and Sriyani) independently reviewed each study using the critical appraisal skills program (CASP) for cohort studies to assess the risk bias. The CASP for cohort studies scores ranged from 0 to 12 [22]. The total appraisal scores of each study ranged from 8 to 12 (highest quality). If a study had a score below 8 points, it was considered to be a high risk of bias. Then, we rechecked the study quality together. If there were conflicting opinions, the study quality score was determined following a joint discussion with the fourth author. “

Point 8: Table 1 – column 6 is unclear – are the patients off / on ART or with / without HZ?

Response 8: Page 4, Table 1

“column 6 are the patients off (without) / on (with) ART” 

Point 9: Table 1 – does incidence refer to overall or post-HAART incidence of herpes zoster?

Response 9: “post-ART incidence of HZ”

Point 10: Results

187: How do the authors explain their finding that higher income and no history of AIDS are risk factors for herpes zoster?

Response 10: Page 5, Line 187 to 192

“Among the 10 studies, patients with high income (pooled RR, 2.64; 95% CI, 1.62–3.65) had the pooled HZ incidence greater than those with low income (pooled RR, 1.33; 95% CI, -0.56–3.22). Among the 4 studies, patients without AIDS history (pooled RR, 0.60; 95% CI, 0.46–0.72) had the pooled HZ incidence greater than those who had AIDS history (pooled RR, 0.40; 95% CI, 0.28–0.54).”

Point 11: 188: Age is a significant risk factor for herpes zoster and most studies reporting on herpes zoster incidence stratify by age. The authors need to include age in their subgroup analysis.

Response 11: Page 5, Line 185 to 187

  1. Due to high heterogeneity, subgroup analyses were conducted and stratified by gender, income, AIDS history, study observational years, HIV risk factors, and CD4 cell count (Table 2).

“Age is the median age in the 11 studies. Not stratify by age, only by study observational years. “

Unlike general population, the age of HIV population infected HZ was much younger. All the 11 studies, the HIV population mean age from 32-41.5 years, the range is below 10, so it is difficult to do the stratification.

Point 12: Table 2: it is unclear in this subgroup analysis if all subjects are in the post-ART area since all studies include a pre / post ART group by inclusion criteria

Response 12:  Mainly, we want compare the incidence of HZ in HIV-infected patients among post-ART era. So we want to focus the post-ART HZ incidence in 11 studies. Overall, 5 studies reported the post-ART HZ incidence, and only 6 of these studies compare pre/post-ART HZ incidence ratios, and do the subgroup analysis.

Page 11, Line 325 to 329

“Fourth, we wanted to compare the incidence of HZ in HIV-infected patients among post-ART era mainly. All the 11 studies reported the HZ incidence among post-ART era, and only 6 studies compared the HZ incidence with pre-ART and post-ART, so we could not determine the effect of the ART completely. “

Point 13:  Discussion

253: How was this pooled incidence calculated? Does it include only patients on ART from each study? What was the methodology for calculating pooled incidence in the 5 studies that did not compare pre / post ART incidences? How does incidence compare to pre-ART era?

Response 13: Mainly, we want compare the incidence of HZ in HIV-infected patients among post-ART era. So we want to focus the post-ART HZ incidence in 11 studies. Overall, 5 studies reported the post-ART HZ incidence, and only 6 of these studies compare pre/post-ART HZ incidence ratios, and do the subgroup analysis.

Page 11, Line 325 to 329

“Fourth, we wanted to compare the incidence of HZ in HIV-infected patients among post-ART era mainly. All the 11 studies reported the HZ incidence among post-ART era, and only 6 studies compared the HZ incidence with pre-ART and post-ART, so we could not determine the effect of the ART completely. “

Point 14: 278: What does this study add by including European countries? What are the expected or hypothesized differences in HZ incidence in this population compared to previous meta-analysis

Response 14: Because seldom studies focus the issue in European countries, we want to compare all the countries data as the global epidemiology data.

Page 10, Line 282 to 287

“A limitation of that study was that it assessed 14 studies for the HZ incidence, but the countries were from Sub-Saharan Africa (one RCT and seven cohort studies), Asia (four cohort studies), and Latin America (two cohort studies); it did not include European countries. Conversely, an advantage of our study was that we assessed the geographic incidence of HZ in HIV-infected populations, include the different countries and race. ”

Point 15: 288: HZ is not caused by herpes simplex virus and is not sexually transmitted

Response 15: Page 10, Line 303 to 306

“HZ is a transmitted disease caused by the herpes simplex virus-2; therefore, HZ incidence, like MSM, is likely to be higher in HIV-infected patients as they are a sexually active population, that are associated with similar risk factors (e.g. sex, partner change rate, condom use). [10,27,28,29].“

Point 16: 318: How does study distinguish between presumptive and conclusive diagnosis of HZ? Add this to methods section.

Response 16: Page 2, Line 94 to 96

“(e) presumptive diagnosis of HZ were as a rash through physician’s examination and diagnosis, and conclusive diagnosis were extracted from ICD codes: 53.0-53.9 (ICD-9) and B02.0-B02.9 (ICD-10).“

Point 17: Conclusions

What recommendations do you have regarding prevention and interventions and for which populations?

Response 17: Page 11, Line 352 to 359

“Clinical management of individuals exposed to varicella-zoster virus should take into consideration the type of exposure, evidence of immunity, and host-immune status with regard to ability to receive varicella vaccination safely. The Shingrix vaccine was immunogenic and had a clinically acceptable safety profile in HIV-infected adults in London, Berlin, Germany and Belgium [32], but most countries not used in HIV-infected population duo to out-of-pocket. This data will be used for health policy makers to determine and prevention programs for people at risk of herpes zoster among those living with HIV.“

Round 2

Reviewer 2 Report

Line 29: definition of post-ART era is unclear - is it a certain year? how is it defined in the various studies?

Line 99: if the inclusion criteria includes studies that have a pre and post ART group, then all 11 studies must have a pre and post ART group. Do the 5 studies that have a post ART incidence only, also have a pre-ART group. If the 5 studies do not, these studies should not be included in the analysis. 

Table 1 – why does Domingo et al. have cases with and without HZ in column 6? This is different from the rest of the studies. 

187: How do the authors explain their finding that higher income and no history of AIDS are risk factors for herpes zoster? This should be addressed in the discussion - provide an analysis of the finding. 

366 Explain in your discussion how the incidence in European counties differs from previously reported global data

303-306: HZ is not caused by herpes simplex virus and is not sexually transmitted. This statement is false.

352-359: english needs to be revised. The author has not sufficiently interpreted their results (the overall incidence etc) in the discussion which prevents them for making a recommendation to vaccinate or not vaccinate HIV infected individuals. The author needs to explain whether incidence is high enough to warrant vaccination. 

Author Response

Response to Reviewer 2 Comments

Point 1: Line 29: definition of post-ART era is unclear - is it a certain year? how is it defined in the various studies?

Response 1:

1. Page 1, Line 29

The definition of post-ART era is “after the introduction of HAART in 1997”

2. The time cut-off point for the comparative study of pre-ART era and post-ART era was in 1997 (https://www.medscape.com/viewarticle/547646). The ART administration protocol in the post-ART era are based on the HAART treatment guidelines of different countries. 

Point 2: Line 99: if the inclusion criteria includes studies that have a pre and post ART group, then all 11 studies must have a pre and post ART group. Do the 5 studies that have a post ART incidence only, also have a pre-ART group. If the 5 studies do not, these studies should not be included in the analysis. 

Response 2: 

Page 2, Line 89 to 91

“(b) studies that included compare the pre-ART (before ART) group and the post-ART (after ART) group or a post-ART (after ART) that utilized ART, regardless of the type, dose, and route of administration”

Page 10, Line 268 to 275

“In our analysis, there were 5 studies that have the post-ART HZ incidence only, and the other 6 studies compare pre-ART and post-ART HZ incidence. The first phase of ART therapy era was from 1997 to 2006, after which the protocol of ART medications was continuously updated. After 2010, the latest studies mainly focus on the post-ART era. Meta-analysis study required more research data for conducting empirical explanations, increasing the sample size, and doing the subgroup analysis. If we exclude the 5 studies, that may increase the research bias and heterogeneity. Finally, early studies may have higher heterogeneity, if there were newer studies, subgroup analysis could be carried out. ”

Point 3: Table 1 – why does Domingo et al. have cases with and without HZ in column 6? This is different from the rest of the studies. 

Response 3: We add the data “0/316” and revised the Table1 on Page 4.

Point 4: 187: How do the authors explain their finding that higher income and no history of AIDS are risk factors for herpes zoster? This should be addressed in the discussion - provide an analysis of the finding.

Response 4: Page 10, Line 276 to 293

“Our finding showed that patients with high income had the pooled HZ incidence greater than those with low income. Accumulating evidence suggests responses to HIV that combine individual level interventions with those that address structural or contextual factors (like housing, income etc.) that influence risks, access to care and health outcomes [24]. Socioeconomic status often determines access to HIV treatment. In resource-poor countries, poverty may prevent access to health care and subsequent treatment; while in resource-rich countries, easy access to care, diagnosis and treatment may early diagnosis of HZ for PLWH [25]. The impact of the introduction of ART on the incidence and mortality of HIV–associated opportunistic infections (OIs) has been well documented in high-income countries (HICs). During the first year of ART, the risk of herpes zoster declined to 2.3%. There was a major reduction in risk for most opportunistic infections with ART use in low- and middle- income countries (LMICs), with the greatest effect seen in the first year of ART treatment [7]. Our finding showed that patients without AIDS history had the pooled HZ incidence greater than those who had AIDS history. Clinically, patients without history of AIDS that come to hospitals for treatment of HZ, and AIDS was concurrent diagnosed with HZ. Herpes zoster was considered to be an early manifestation of HIV-1 infection, and even a harbinger of the AIDS as immunosuppressive conditions [26]. ”

Point 5: 366 Explain in your discussion how the incidence in European counties differs from previously reported global data

Response 5: Page 10 to 11, Line 311 to 317

“A systematic literature review showed that annual HZ incidence of general population throughout Europe, varying by country from 2.0 to 4.6/1000 person-years with no clearly observed geographic trend [29]. HZ incidence increases with age, and quite drastically after 50 years of age; incidence rates were higher among women than men, and this difference increased with age. In the European (WHO) database, the overall mortality ranged from 0 to > 0.07/100,000. The age- and gender-specific HZ mortality rates from the other databases showed that while in younger age groups the HZ mortality rate was higher in males, in older patients the rate was much higher in women [32]. The HZ incidence was higher in HIV-infected population among pre-ART era than general population [4,14].”

Point 6: 303-306: HZ is not caused by herpes simplex virus and is not sexually transmitted. This statement is false.

Response 6Page 11, Line 332

“HZ is a transmitted disease caused by the varicella zoster virus”

Point 7: 352-359: english needs to be revised. The author has not sufficiently interpreted their results (the overall incidence etc) in the discussion which prevents them for making a recommendation to vaccinate or not vaccinate HIV infected individuals. The author needs to explain whether incidence is high enough to warrant vaccination. 

Response 7: Page 11, Line 343 to 354

“Currently, there are two vaccines on the market, Zostavax (a live vaccine) and Shingrix (a recombinant zoster vaccine). Recommendation for vaccination depends on HIV infected individuals’ immunity, suggested for CD4+ Cell Counts >200 Cells/mL after ART use. Zostavax is less effective in older individuals and is contraindicated in immunosuppressive conditions (HIV/AIDS), during immunosuppressive drug therapy and pregnancy [36]. A randomized, double-blind, controlled trial in HIV-infected adults with CD4+ cell counts >200 cells/mL virally suppressed on ART, the results showed that two doses of Zostavax in HIV-infected adults suppressed on ART with CD4+ counts ≥200 cells/µL were safe and immunogenic [37]. The Shingrix vaccine was immunogenic and had a clinically acceptable safety profile in HIV-infected adults in London, Berlin, Germany and Belgium [38], but most countries not used in HIV-infected population due to out-of-pocket.”

Round 3

Reviewer 2 Report

Comments have been addressed.